

# Male foraging efficiency, but not male problem-solving performance, influences female mating preferences in zebra finches

Véronique Chantal, Julie Gibelli and Frédérique Dubois

Département de sciences biologiques, Université de Montréal, Montréal, Canada

## ABSTRACT

Experimental evidence suggests that females would prefer males with better cognitive abilities as mates. However, little is known about the traits reflecting enhanced cognitive skills on which females might base their mate-choice decisions. In particular, it has been suggested that male foraging performance could be used as an indicator of cognitive capacity, but convincing evidence for this hypothesis is still lacking. In the present study, we investigated whether female zebra finches (*Taeniopygia guttata*) modify their mating preferences after having observed the performance of males on a problem-solving task. Specifically, we measured the females' preferences between two males once before and once after an observation period, during which their initially preferred male was incapable of solving the task contrary to their initially less-preferred male. We also conducted a control treatment to test whether the shift in female preferences was attributable to differences between the two stimulus males in their foraging efficiency. Finally, we assessed each bird's performance in a color associative task to check whether females can discriminate among males based on their learning speed. We found that females significantly increased their preference toward the most efficient male in both treatments. Yet, there was no difference between the two treatments and we found no evidence that females assess male cognitive ability indirectly via morphological traits. Thus, our results suggest that females would not use the males' problem-solving performance as an indicator of general cognitive ability to gain indirect fitness benefits (*i.e.*, good genes) but rather to assess their foraging efficiency and gain direct benefits.

## INTRODUCTION

As the brain structures needed to acquire, process, store and use information from the environment are costly to develop and maintain, cognitive abilities in both humans and animals are often considered as an honest indicator of genetic quality that should be used as a mate-choice criterion (*Jacobs, 1996*; *Miller, 2000*; *Boogert, Fawcett & Lefebvre, 2011*). More precisely, improved cognitive abilities can help animals to respond quickly and adequately to environmental changes (*Kotrschal & Taborsky, 2010*). Females, therefore, might benefit from choosing a mate with higher cognitive ability because it

Corresponding author
Frédérique Dubois,
frederique.dubois@umontreal.ca

would be better to cope with changing conditions, hence providing them and their offspring with better resources. Females could also gain indirect benefits when the cognitive traits are heritable (*Croston et al., 2015*). Supporting the idea that males with better cognitive skills are preferred as mates, two studies have demonstrated that males with better spatial learning abilities are more attractive to females in both meadow voles (*Microtus pennsylvanicus*) (*Spritzer, Meikle & Solomon, 2005*) and guppies (*Poecilia reticulata*) (*Shohet & Watt, 2009*). Also there is good evidence that birdsong, which is an indicator of brain development (*Farrell, Kriengwatana & MacDougall-Shackleton, 2015*), plays an important role in mate attraction (*Searcy & Andersson, 1986*; *Nowicki, Searcy & Peters, 2002*). Yet, relatively few studies except those concerning song learning in birds, have looked at how individuals assess the cognitive capacity of the opposite sex. Consequently, little is known about the traits reflecting enhanced cognitive skills on which females might base their mate-choice decisions in other taxa or even in bird species in which song complexity is not a meaningful indicator of cognitive capacity (*Boogert et al., 2011*; *Templeton, Laland & Boogert, 2014*).

Several authors have suggested that male foraging performance could be such a cue that females would use as an indicator of cognitive capacity (*Boogert, Fawcett & Lefebvre, 2011*). In particular, females could discriminate among males based on their ability to solve novel problems. Indeed, experimental evidence has shown that individuals of the same population may differ widely in their problem-solving success and that this trait correlates positively with performance on various learning tasks (*Bouchard, Goodyer & Lefebvre, 2007*; *Boogert, Giraldeau & Lefebvre, 2008*; *Cole, Cram & Quinn, 2011*; *Overington et al., 2011*; *Aplin, Sheldon & Morand-Ferron, 2013*; *Griffin et al., 2013*; *Templeton, Laland & Boogert, 2014*; *Shaw et al., 2015*). These findings suggest that males with better problem-solving ability would have higher general cognitive ability (*Shaw et al., 2015*). In addition, recent research has established a link between problem-solving ability and mating success, hence providing evidence that females would also obtain direct fitness benefits from choosing mates with better cognitive skills. Specifically, *Keagy, Savard & Borgia (2009)* and *Keagy, Savard & Borgia (2011)* have reported that male satin bowerbirds (*Ptilonorhynchus violaceus*) with better problem-solver ability in the field obtain more copulations, while two recent studies on great tits (*Parus major*) have demonstrated that more cognitively skilled mates that are faster problem solvers produce more offspring (*Cole et al., 2012*; *Cauchard et al., 2013*).

To date, however, evidence for the hypothesis that females use male foraging performance as an indicator of cognitive ability is indirect. Indeed, several studies in birds (*Hill, 1990*) and fish (*Pike et al., 2007*) have reported that females prefer brighter or more colored males, probably because they are more efficient in acquiring food and hence ingest more carotenoids responsible for brightly colored sexual ornaments. Yet, it is unclear whether carotenoid coloration reflects male foraging success and whether females use direct observation of male cognitive performance rather than traits that are correlated with cognition when choosing a mate. Only *Snowberg & Benkman (2009)* have demonstrated that female crossbills (*Loxi curvirostra*) rely on male foraging performance to choose a mate. More precisely, they found that females

that had observed two males that differed in their feeding rate preferred the most efficient one. However, there is no evidence that male crossbills that are more efficient at extracting seeds from conifer cones have better cognitive skills. Therefore, no study has yet directly tested whether females discriminate among males through direct observation of their performance on a foraging task that indicates cognitively ability.

In the present study, we addressed this question by investigating whether female zebra finches (*Taeniopygia guttata*) modify their mating preferences after having observed the foraging performance of males on a problem-solving task. Although male song has been found to be important for female choice in this species (*Riebel, 2009*), recent findings indicate that song complexity would not be a good indicator of general cognitive ability (*Templeton, Laland & Boogert, 2014*), as previously thought (*Boogert, Giraldeau & Lefebvre, 2008*). Females, therefore, might benefit from using other cues that best reflect a male's overall cognitive ability, such as its ability to solve novel problems. Thus, to assess the influence of this cue on female mate-choice decisions, we trained males to solve a task, and then we measured the mating preferences of each female twice: before and after she had observed the performance of two stimulus males on the task (main treatment). We experimentally manipulated the performance of the two males during the observation period, so that each female could observe her initially preferred male that was incapable of solving the task (*i.e.*, the non solver) and her initially less-preferred male (*i.e.*, the solver) that, on the contrary, was highly efficient at solving the task. Furthermore, because only the solver could access food, we conducted a control treatment to test whether the change in females' preferences observed in the main treatment could be explained by differences among males in their foraging efficiency rather than in their ability to solve the task. Finally, we measured each bird's learning performance in a color associative task in order to check 1) whether females, prior to the observation period, could discriminate between the two males based on their learning performance and hence preferred the male that learned faster, and 2) whether the ability of females to assess male cognitive ability was related to their own learning performance.

## METHODS

### Subjects and housing

We used 40 (30 females and 10 males) commercially purchased unrelated adult zebra finches obtained from a local breeder (Exotic Wings & Pet Things, St Clements, Ontario, Canada). Twenty-two birds (18 females and 4 males) and 18 birds (12 females and 6 males) were used in the main and control treatments, respectively. Outside the testing periods, the birds were kept in groups of two or three in same-sex cages ($10 \times 40 \times 30$ cm) with a 14:10 h light: dark photoperiod at approximately $23 \pm 1\,°C$. They had ad libitum access to seeds, water and cuttlefish bone. In addition, their diet was supplemented once a week with egg yolk mixture and vegetables. The experiments described in this study were approved by the Animal Care Committee of the University of Montreal (animal care permit #14-073) and conformed to all guidelines of the Canadian Council on Animal Care.

## Main treatment

### Problem-solving task

Before we measured the preferences of each female between one solver and one non solver, we trained the males to solve a task, which consisted of a transparent plastic tube filled with millet seeds and closed with a lid that the bird had to flip to get access to the food (Fig. 1). Training sessions occurred between 7 and 13h00 after overnight food deprivation and lasted for 20 consecutive days with two sessions per day separated by 3 h. Males were trained by pairs in their housing cage that was divided by an opaque partition in two sections. Thus, the birds could not scrounge food or observe each other's behavior. The day before the training began, we placed two apparati outside of the cage, to allow the birds to become familiar with them. Then the training procedure consisted of the following three steps: 1) we provided the birds with an open tube (*i.e.*, with no lid); 2) once the birds had eaten for 10 s in step 1, a lid was just deposited on the tube, so that the birds could easily get access to the food by pushing down the lid. An individual who succeeded in pushing down the lid had access to the food during 10 s before the lid was replaced; 3) once the birds had succeeded five times in step 2, the lid was pressed halfway so that the birds had now to flip the lid to get access to the food. The training was over when the birds could open the tube at least 10 times during a 60 min period.

### Mate-choice apparatus and experimental procedure

We measured female mating preferences with a classical binary choice apparatus (Fig. 2) that comprised three compartments: A) the observation compartment where the focal female could see both males simultaneously, B) the choice compartment where she could see only one stimulus male at a time and C) the male compartment divided into two identical chambers, each housing a single male. Before the beginning of the experiment, males and females were placed individually in the apparatus during one hour for 10 days to become familiar with their environment. Then we measured each female's preference twice (*i.e.*, initial and final preferences): before and after an observation period, during which she could observe one of the two stimulus males solving the task while the other did not.

The initial preference of each focal female was measured following this procedure: after the two stimulus males had been placed in the male compartments, we introduced the test female in the observation compartment and after a 15 min period, we gently lifted the transparent partition between the observation and choice compartments. We then measured the time she spent on the perches in the neutral zone and in front of each male during two consecutive periods of 30 min each, switching the position of the males after 30 min. To control for differences in the stimulus males' songs, we masked their songs during the duration of the mate choice tests by playing a recorded chorus of calls and songs from male and female zebra finches. Furthermore, to ensure that the females were able to distinguish between the two males, we formed the pairs so that the two stimulus males differed in terms of size, plumage and beak color.

The five days following the initial preference test, each female was placed in the observation compartment for two periods per day during which she could observe the two

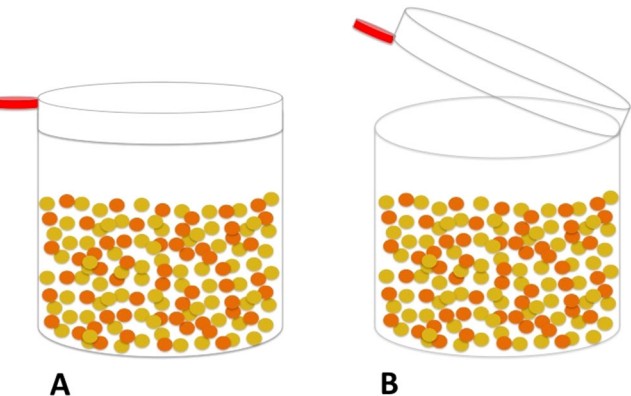

**Figure 1 Side view of the motor learning task.** The lid of the plastic tube was pressed either halfway to allow the bird to easily flip the lid or fully pressed to prevent the bird to get access to the food.

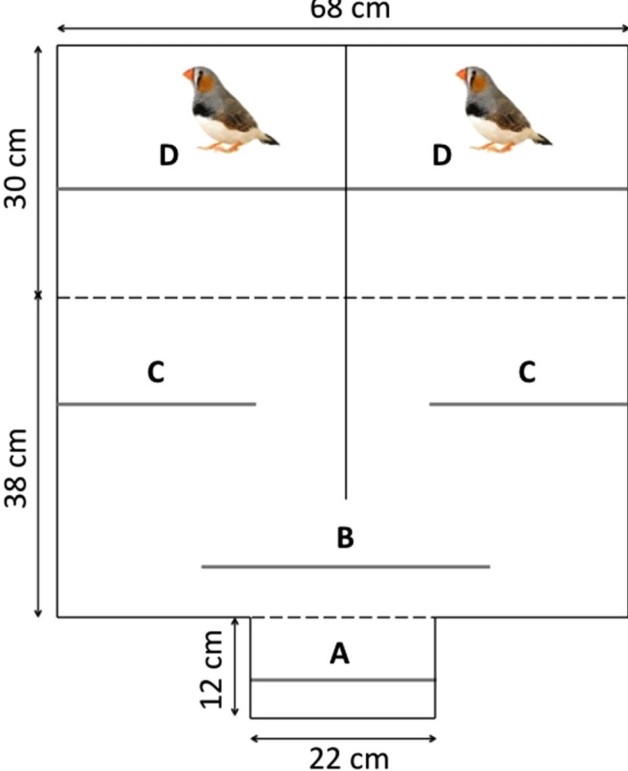

**Figure 2 Top view of the mate-choice apparatus with: the observation compartment (A), the male compartment (D) and the choice compartment divided into the neutral zone (B) and the choice zone (C).** The grey lines represent the perches while the black lines correspond to the partitions that were opaque (full lines) or clear (dashed lines).

stimulus males while they were interacting with the problem-solving task. Before each observation period, the two stimulus males were food deprived for 3 h. Then, in order to manipulate their success, one of them (*i.e.*, the solver) was provided with a tube the lid of which was pressed only halfway and hence that could be easily opened, while the other male (*i.e.*, the nonsolver) was provided with a tube the lid of which was fully pressed and

hence that was impossible to open. For each female, the easy task was provided to her initially less-preferred male while the difficult task was provided to her initially preferred male. Each observation period ended after 60 min or once the most efficient male had solved the task 10 consecutive times, whichever occurred first. In addition, to prevent female-male interactions and other distractions during the observation periods, we placed three natural-spectrum, 60 W light bulbs directly in front of each male's compartment, making it harder for the males in their brightly lit environment to detect the female in her shaded compartment in front of them.

After 24 h, we measured the final preference of the focal female using the same procedure as for the initial tests. All females, except one female that was injured after the initial test, were tested for their final preferences. In addition, although females observed the performance of the males on the problem-solving task only during the observation sessions, males were provided with the task every day during habituation and testing periods, so that they do not forget how to flip the lid.

## Control treatment

We used exactly the same procedure as described above, except for the observation period, during which the test female could observe twice a day the two stimulus males while they were searching for seeds within a dish (13 × 7 × 3.5 cm). The food dishes that were provided to the two stimulus males both contained a double layer of dried peas that acted as obstacles, thereby forcing the birds to move them around to detect and gain access to the millet seeds when they were present. In addition, in order to manipulate the feeding rate of the two stimulus males, the initially preferred male was provided a dish that contained no seeds while the other male was provided a dish with 30 millet seeds. Each observation period ended after 10 or 2 min after the most efficient male had stopped searching for food, whichever occurred first.

## Associative learning task

We measured the performance of all individuals (*i.e.*, both males and females) in a color associative task as the number of trials needed to find six consecutive times the rewarded feeder. Specifically, the birds were tested individually in an experimental apparatus that comprised an observation chamber (20 × 50 × 30 cm) and a choice chamber (40 × 50 × 30 cm) that were separated from each other by a transparent removable partition. The choice chamber was divided into four symmetrical corridors, and at the end of each corridor we placed four white feeders that were positioned in front of four colored dots (*i.e.*, yellow, cyan, pink and black) whose position changed randomly from one trial to the next. The rewarded feeder (*i.e.*, the feeder placed in front of the yellow dot) always contained four millet seeds, while the other feeders were empty.

Prior to testing, we trained the birds to eat from the feeder that was deposited within their home cages. Then, the birds were placed in the experimental apparatus to become familiarized with the environment. They spent at least 3 h per day for two weeks in the apparatus until they could explore the four corridors and eat without fear from the feeders, whatever their position.

Before each testing day, the birds were food deprived for 3 h. They experienced a maximum of 25 trials per day during four consecutive days or until they had reached the learning criterion, whichever occurred first.

At the beginning of each trial, the bird was confined in the observation chamber for 2 min. Then, the observer gently lifted the removable partition, thereby allowing the bird to enter in the choice chamber and choose one of the four feeders. Once the bird had chosen a corridor, we noted whether it had succeeded or failed. If the bird had succeeded, it could eat the four seeds before returning to the observation chamber. On the contrary, if the bird had failed, the observer either gently activated the removable partition to encourage the bird to return into the observation chamber if it had obtained food during the previous trial or let it explore the other corridors and find the rewarded feeder otherwise. Such a procedure was adopted to insure that all the birds ate approximately the same amount of food during each session and that differences among individuals in their learning speed, therefore, were not due to differences in their level of satiety. All but three injured birds (two males and one female) were used for this experiment.

## Statistical analyses

To determine whether the females were capable of discriminating between the two stimulus males based on their learning capacity, we tested whether the percentage of time spent in front of the male who resolved the color association task faster was significantly larger than 50% using a one-sample t-test. Because we used five different pairs of males for the preference tests, we also conducted a one-way ANOVA to assess whether female preferences differed among the pairs of stimulus males. Next, we compared the average learning performance of the females that expressed a marked preference (*i.e.*, spent 55% or more of their choosing time in front of one male) for either the fast or the slow learning male using a t-test, and we used a Pearson correlation coefficient to determine whether the relative time spent by females in front of their initially preferred male was correlated with the difference between the two stimulus males in their learning speed.

For both treatments, we assessed whether the change in the females' preferences (*i.e.*, the relative time spent in front of the most efficient male in the final preference test minus the relative time spent in front of the same male in the initial preference test) significantly differed from zero using a paired t-test, and then we performed a t-test to determine if the change in preferences differed between the two treatments. We also verified that the relative time spent in the choosing zone was not significantly different between the initial and final test preferences using a paired t-test, and for both variables (*i.e.*, change in the females' preferences and change in their relative time spent in the choice zone) we conducted a one-way ANOVA to test for an effect of pair identity. Finally, we used Pearson's correlation coefficient to test for an association between the change in females' preferences and their learning score. Data were excluded from the analyses when females spent less than 30% of their time in the choosing zone. Statistical analyses were done with SPSS 23.0 for Mac.

## RESULTS

During the initial preference test, females on average (X ± SE) spent 47.83 ± 4.60% of their choosing time in front of the faster learner of the two stimulus males in the color association task, which is not significantly different from 50% ($t_{23}$ = −0.471, $P$ = 0.642). Furthermore, there was no significant effect of the identity of the stimulus males on the expression of female preferences ($F_{3,20}$ = 0.066, $P$ = 0.977). Female choice, therefore, was random with respect to male learning performance in the color association task. The relative time spent by females in front of their less-preferred male was not correlated either with the difference in learning speed between the two stimulus males ($r$ = −0.075, N = 30, $P$ = 0.694). This finding indicates that females that had to choose between two males that differed largely in their learning performance were not more likely to prefer the faster learner of the two stimulus males than those that had to choose between two potential mates with more similar learning speeds. Finally, the mean number of trials needed to solve the color associative learning task was not significantly different between females that preferred the faster learner and those that preferred the slower learner of the two males ($t_{10}$ = −0.622, $P$ = 0.548).

The time spent by females in the choice zone was not significantly different between the initial and final preference tests ($t_{27}$ = −0.335, $P$ = 0.740). On the contrary, we found that females significantly increased their preference toward the initial less-preferred male after having observed the performance of the two stimulus males in both treatments (main treatment: $t_{15}$ = 2.608, $P$ = 0.020; control treatment: $t_{11}$ = 2.472, $P$ = 0.031; Fig. 3). Yet, there was no significant effect of the treatment on the shift in female preferences ($t_{26}$ = 1.164, $P$ = 0.255) and neither variable was affected by the identity of the stimulus males (change in the relative time spent in the choice zone: $F_{4,24}$ = 0.072, $P$ = 0.990; change in the relative choosing time spent in front of the initially less-preferred male: $F_{4,23}$ = 0.832, $P$ = 0.579). Finally, we found no correlation between the females' learning speed in the color association task and the magnitude of the change in their mating preferences in the main treatment ($r$ = 0.178, N = 16, $P$ = 0.509) or in the control treatment ($r$ = 0.269, N = 11, $P$ = 0.424).

## DISCUSSION

We found that zebra finch females significantly increased their mating preference toward the most efficient (initially less preferred) male, after having observed the performance of the two stimulus males in both treatments. Because in both treatments, the two stimulus males differed in their feeding rate, our results suggest that females use male foraging efficiency as a mate-choice criterion. This result is in agreement with the study of *Snowberg & Benkman (2009)* who reported that red crossbill females also preferred the male that was the more efficient forager. In zebra finches, variation among individuals in their feeding rate causes variation in their reproductive success (*Lemon & Barth, 1992*; *Lemon, 1993*). More precisely, because individuals with high rates of energy gain have more time and energy available for reproduction compared with less efficient foragers, they are able to produce more offspring that also survive better. Female zebra

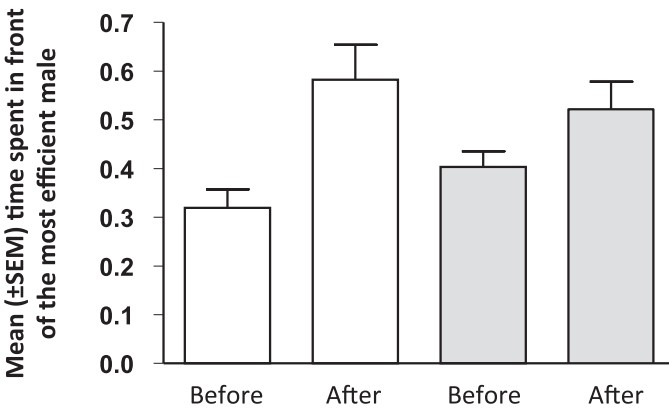

**Figure 3 Mean (± SEM) percent of choosing time spent in front of the male that was the most efficient forager, before and after females had observed the males' performance in the main (white bars) and control (grey bars) treatments.**

finches, therefore, can gain direct fitness benefits from choosing mates based on their foraging efficiency.

Yet, contrary to our expectations, we found no support for the hypothesis that zebra finch females discriminate among potential mates based on their problem-solving ability. Indeed, we detected no significant difference in the change of female preferences between the two treatments, which means that the capacity of the males to solve the task in the main treatment was unimportant for females compared to the males' feeding rate. Thus, our results indicate that female zebra finches do not use male problem-solving performance as an indicator of cognitive capacity. One reason that could explain this finding is that mate assessment based on male foraging performance likely requires considerable time, which would prevent most females from using this trait as a mate-choice criterion. Indeed, as zebra finches are opportunistic breeders, starting to breed immediately after rain (*Zann, 1996*), females have to make quick mating decisions. Under natural conditions, however, the probability of observing a cognitively demanding foraging behavior (*e.g.*, an innovation) is expected to be very low. In order to reduce the cost of mate assessment, females would then benefit from using morphological traits that are correlated with cognitive abilities, instead of assessing directly the males' cognitive performance. However, we found no evidence for this explanation.

Indeed, prior to the observation of the males' performance, females did not prefer the faster learner of the two stimulus males and we found no evidence, either, that they chose assortatively based on learning capacity. These findings indicate that female zebra finches do not assess male cognitive ability indirectly via morphological traits or courtship displays, irrespective of their own cognitive abilities. Although it is possible that we failed to detect a preference of females for the faster learner of the two stimulus males because there was not enough variation among them in their learning performance, this explanation is unlikely. Indeed, we found no correlation between the strength of female preferences and the difference in learning speeds between the two stimulus males, which means that the time spent by females in front of the fast-learning male was not

influenced by the amount of variation between the two potential mates in their cognitive ability. So, our results suggest that females would not use the males' performance on different learning tasks as an indicator of general cognitive ability to gain indirect fitness benefits (*i.e.*, good genes) but rather to assess their foraging efficiency and hence gain direct fitness benefits. This conclusion is supported by the fact that several authors have reported non-significant correlations among individual performance on different cognitive tasks (*Boogert et al., 2011*; *Templeton, Laland & Boogert, 2014*; *Farrell, Kriengwatana & MacDougall-Shackleton, 2015*; *Kriengwatana et al., 2015*), which strongly suggests that different cognitive measures would each reflect a specific ability. As a consequence, though our results need to be further confirmed, we argue that cognitive traits could evolve through sexual selection only if enhanced cognitive skills enable males to acquire more resources and hence to produce more viable offspring.

In conclusion, our results showed that female zebra finches use direct observation of foraging efficiency to guide their mate-choice decisions, probably because females mated with highly efficient foragers are able to produce more offspring that survive better. Yet, we found no evidence that females assess males' cognitively ability either directly via observation of their performance on a problem-solving task or indirectly via morphological traits that are correlated with their learning ability. Thus, our results do not support the hypothesis that female zebra finches would use male learning ability as an indicator of general cognitive ability, but additional studies would be required to confirm our conclusions. In particular, given that male song is an important mate-choice criterion used by females, future studies should explore whether song advertises direct benefits, indirect benefits or both (*Farrell, Kriengwatana & MacDougall-Shackleton, 2015*).

## ACKNOWLEDGEMENTS

We thank Guillaume Pilon for his help during data collection. We are also grateful to Academic Editor Clint Kelly, Scott MacDougall-Shackleton and an anonymous reviewer for providing constructive suggestions and comments.

### Funding

During this study, V. Chantal was supported by a research grant (research grant #R0013057) awarded to F. Dubois by the Natural Sciences and Engineering Research Council of Canada (NSERC) while J. Gibelli received financial support through a scholarship from the Principauté de Monaco. The funders had no role in study design, data collection and analysis, decision to publish, or preparation of the manuscript.

### Grant Disclosures

The following grant information was disclosed by the authors:
Natural Sciences and Engineering Research Council of Canada (NSERC): #R0013057.

## Competing Interests

The authors declare that they have no competing interests.

## Author Contributions

- Véronique Chantal conceived and designed the experiments, performed the experiments, analyzed the data, wrote the paper, prepared figures and/or tables, reviewed drafts of the paper.
- Julie Gibelli conceived and designed the experiments, performed the experiments, wrote the paper, reviewed drafts of the paper.
- Frédérique Dubois conceived and designed the experiments, analyzed the data, wrote the paper, prepared figures and/or tables, reviewed drafts of the paper.

## Animal Ethics

The following information was supplied relating to ethical approvals (i.e., approving body and any reference numbers):

The experiments described in this study were approved by the Animal Care Committee of the University of Montreal (animal care permit #14-073) and conformed to all guidelines of the Canadian Council on Animal Care.

## Data Deposition

The raw data has been supplied as Supplemental Dataset Files.

## Supplemental Information

Supplemental information for this article can be found online at http://dx.doi.org/10.7717/peerj.2409#supplemental-information.

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
