# Peer review of "Male foraging efficiency, but not male problem-solving performance, influences female mating preferences in zebra finches"

_PeerJ, doi:10.7717/peerj.2409_

## Round 0.1 · original submission · Major Revisions

The 2 reviewers have touched on the more salient aspects of your methodology and data-interpretation and so I have nothing further to add on that front. Here, however, I will offer some editorial suggestions that will hopefully improve your manuscript.

Line 71: “in acquiring”

Line 89: delete “the”

Line 110: replace “into” with “in”

Line 112: I don’t understand this: “wearing”

Line 120: Delete: “problem-solving”

Line 125: Change “other” to “other’s”

Line 126: Change “begun” to “began”

Line 126: Change “apparatuses” to “apparati”

Line 152: Delete “all”

Line 162: Delete “that”

Line 167: Change “distraction” to “distractions”

Line 172: Change “expect” to “except”

Line 190: Change “to” to “from”

Line 241: Change “preferences” to “preference”

> Line 233: in females’ preferences XXXXXX

Line 240: Should be “random with respect…”

Line 265 and elsewhere: Change “variations” to “variation”

Line 265: Change “cause” to “causes”

Line 261: Delete “then”

Line 262: “Suggest” rather than “demonstrate” might be a better choice here.

Line 279-280: Change “to use” to “from using”

Line 285: Perhaps this would be a better choice: “However, we found no evidence to support this hypothesis.” This would reduce the use of the word “Yet”.

Line 287: Change “preferred” to “prefer”

Line 291: Change “of” to “by”

Line 295: Insert hyphen between “fast” and “learning”

Reviewer 1 ·

Basic reporting

The authors have investigated an interesting hypothesis and they have designed experiments that are probably adequate enough to reach some interesting conclusions. However, their reporting of the Results is minimal and incomplete. Moreover, the conclusions reached in the Discussion seem to be largely disconnected from the actual Results, or they are very poorly presented.
More minor points follow:
49-51: Please explain what cognitive abilities were assessed in the studies on guppies and bowerbirds.
64-65: Make it clear what is implied here by ‘higher quality’.

Experimental design

The only scores reported are preference of the females determined as time next to the solver relative to time next to the non-solver. The total time scored as ‘next to’ either bird should also be reported (i.e., time next to solver plus time next to non-solver). Does strength of preference (i.e., relative time) vary with respect to total time?
154-156: Were the physical differences randomized with respect to solvers and non-solvers? This is important and needs to be addressed statistically.

Validity of the findings

Given the minimal reporting of the Results, it is impossible to tell whether the findings are valid or whether they adequately address the aim of the paper. The Results are not reported at all clearly. Where is the statistical evidence in support of the first sentence of the Discussion, for example? The terminology used is neither transparent of consistent. The authors need to decide what exactly is purely a cognitive task and what is a foraging task. They also need to engage with their own results rather than simply lining up with research already reported.
298: This sentence implies that the authors have reported on song learning ability. This is incorrectly expressed.

Additional comments

Corrections required to grammatical expression:
57: Correct ‘such as cue’ to ‘such a cue’.
112: Do you mean ‘wearing’ or ‘weaning’?
125: Correct ‘other’ to ‘other’s’.
161 and 163: Correct ‘whose lid’ to ‘the lid of which’. Also make this correction in 192.
168: Insert ‘it’ after ‘making’.
172: Correct ‘expect’ to except’.
190: Correct ‘separated to’ to ‘separated from’.
240: Correct ‘in respect to’ to ‘with respect to’.
287: Correct ‘preferred’ to ‘prefer’.
Fig. 2: The footnote to this figure does not match the figure. What is compartment C? D does not exist? What does the green colour indicate?
Fig. 3: Please state exactly what is plotted (means, sems?) and what the grey and white bars represent.

·

Basic reporting

Apart from a few typographical errors I believe the basic reporting meets all PeerJ policies.

Experimental design

All good.

Validity of the findings

Generally good, though I have some points on which I would like further clarification (see General Comments)

Additional comments

I found this article cleverly designed, topical, and well-written. However, I think that some of the methods need more clarity.

Line 144-146 is unclear to me -did the males interact with the feeding apparatus during the preference tests? Or only during the intervening observation period? I ask because an alternative explanation is that females are not exhibiting a change in mate preference, but just a food learning association. That is, could females spend more time with the successful problem solvers and more efficient foragers not because they are attracted to those males, but because they just associate those males with food? I am not sure this explanation can be ruled out even if the males were feeding only during the middle observation phase.

It is also unclear to me how males were repeatedly used as stimuli. With 29 females and 10 males it is clear that individual males were re-used, but was each female given a novel pairing of males? Could this contribute to non-independence in the analyses?

Finally, it is not clear how the colour association task performance was scored. The results refer to learning scores, but how this is calculated was not at all clear in the methods.

Minor comments:
line 51-53 seems to dismiss a very large literature. Song learning is clearly a cognitive ability and several studies have looked at correlations of song with other aspects of cognition. Similarly line 80-81 implies that birdsong is not a cognitively demanding behaviour. I would disagree with that as birdsong requires exceptional learned motor control, auditory feedback, and reference to memory.

line 112: "egg yolk wearing mixture" -is this a typo?

line 130: is "topping" a typo?

line 240: learning performance on which task?
line 241: how were these scores calculated (see main comments above)?

line 287: is "preferred" a typo?

line 302: the first reference to mice seems a bit out of place. The topic of correlated performance on cognitive tasks in birds is reviewed quite thoroughly in Farrell et al. 2015, and another reference that addresses this in zebra finches is Kriengwatana et al. in Behaviour 152 (2015) 195–218. I suggest either providing example studies for birds or zebra finches, or citing review articles for statements like this.

Fig 2 legend. The red perch did not appear very red in my pdf file. Please make the neutral perch more distinctive in the figure.

---

## Round 0.2 · accepted · Accept

I have evaluated your revision and rebuttal and thank you very much for addressing mine and the reviewers' comments and suggestions.